# Using Detached Industrial Hemp Leaf Inoculation Assays to Screen for Varietal Susceptibility and Product Efficacy on *Botrytis cinerea*

**DOI:** 10.3390/plants12183278

**Published:** 2023-09-15

**Authors:** Karen Kirkby, Sharlene Roser, Krista Plett

**Affiliations:** 1NSW Department Primary Industries, Australian Cotton Research Institute, 21888 Kamilaroi Highway, Narrabri, NSW 2390, Australia; sharlene.roser82@gmail.com; 2NSW Department Primary Industries, Elizabeth Macarthur Agricultural Institute, Woodbridge Road, Menangle, NSW 2568, Australia; krista.plett@dpi.nsw.gov.au

**Keywords:** *Botrytis cinerea*, *Cannabis sativa*, medicinal cannabis, grey mould, fungal pathogens, fungicide screening, detached leaf assays

## Abstract

In greenhouse production, grey mould caused by *Botrytis cinerea* Pers. is one of the most widespread and damaging diseases affecting medicinal cannabis (MC). Fungicide options to control this disease are extremely limited due to the regulations surrounding fungicides and chemical residues as the product end users are medical patients, often with compromised immune systems. Screening for alternative disease control options, such as biological and organic products, can be time-consuming and costly. Here, we optimise and validate a detached leaf assay as a quick and non-destructive method to evaluate interactions between plants and pathogens, allowing the assessment of potential pathogens’ infectivity and product efficacy. We tested eight industrial hemp varieties for susceptibility to *B. cinerea* infection. Using detached leaves from a susceptible variety, we screened a variety of chemical or organic products for efficacy in controlling the lesion development caused by *B. cinerea*. A consistent reduction in lesion growth was observed using treatments containing Tau-fluvalinate and Myclobutanil, as well as the softer chemical alternatives containing potassium salts. The performance of treatments was pH-dependent, emphasizing the importance of applying them at optimal pH levels to maximise their effectiveness. The detached leaf assay differentiated varietal susceptibility and was an effective method for screening treatment options for diseases caused by Botrytis. The results from the detached leaf assays gave comparable results to responses tested on whole plants.

## 1. Introduction

Plant diseases in medicinal cannabis (MC) and industrial hemp (IH) are a major challenge for commercial production. Both MC and IH belong to *Cannabis sativa* L., but the notable difference between the two is the tetrahydrocannabinol (THC) content. MC contains more than 0.3% THC dry weight, while IH contains 0.3% THC or less. In countries such as Australia, where MC production has been legalised, IH is typically grown in field, while MC is grown under strict licensing conditions in intensive covered cropping facilities (glasshouses). There are over 88 fungal pathogens known to affect MC and IH plants [1], and growing healthy plants in intensive production systems is particularly challenging. Additionally, fungal control options are limited because of the strict regulations surrounding the level of pesticide residues to ensure the quality of MC products. The lack of formal knowledge regarding MC physiology and pathology has led to insufficient disease reduction methods available. Low genetic diversity after repeated intercross-breeding and clonal propagation by vegetative cuttings can lead to increased susceptibility of the plant to pathogens [2]. This is exacerbated by the high-intensity growth conditions for MC compared to fibre-type IH plants [3].

Besides direct yield losses, outbreaks of phytopathogens in MC crops also raise human health concerns. For example, exposure to *Botrytis cinerea* Pers. and *Alternaria alternata* (Fr.) Keissl can cause mould allergies, asthma, and opportunistic infections [1,4]. Microbiological contaminants can cause health problems in MC consumers, especially immunocompromised patients [1], and production workers who are repeatedly exposed to airborne fungal spores [5]. Cultural practices are often insufficient to control MC pathogen outbreaks, managing diseases in intensive covered cropping facilities is particularly challenging because natural antagonists are mostly absent, and the use of pesticides might leave residues on commercialised products [6].

*B. cinerea* (grey mould) is an airborne necrotrophic pathogen that causes symptoms including seedling damping-off, stem canker and inflorescence bud rot. Large and tightly packed inflorescences are a desirable trait for the MC industry but also increase their susceptibility to fungal pathogens, such as *B. cinerea*, that thrive in dense and moist environments. *B. cinerea* is a necrotrophic pathogen unable to extract nutrients from living tissues; therefore, it must first penetrate the host surface before killing and macerating the plant cells to sustain its growth [7]. Consequently, when its airborne conidia land on a suitable host, the conidia need ready access to exogenous nutrients and water on the plant surface to germinate, form appressoria, and promptly initiate infection. This may explain why germinating conidia are highly susceptible to competition, inhibitory substances, and interferences from other microorganisms in the phyllosphere [8]. Postharvest sterilisation methods are effective at lowering grey mould contamination on MC inflorescence, but they are costly, difficult to implement, and cannot replace preventative control [2]. Therefore, the use of biological or organic control options in MC is widely suggested as a suitable alternative to chemical treatments.

Screening for host varietal resistance or susceptibility to pathogen infection and product efficacy using whole plants is time-consuming, requires segregated space within the glasshouse away from healthy plants, and often leads to plant death. Alternatively, screening for varietal resistance/susceptibility and product efficacy using detached leaf assays has the potential to be a fast and reliable alternative to whole plant screening. Detached leaf inoculation assays have been used for screening host resistance to various pathogens in tomatoes [9], strawberries [10], chickpeas [11], and apples [12] but have not been developed for use in MC or IH. For practical reasons surrounding legislation and security requirements for MC grown in Australia, our research on *B. cinerea* control was conducted using IH as a model crop for MC. The aim of this research was to evaluate, standardise, and validate an effective, fast detached leaf inoculation assay to (1) differentiate susceptibility to *B. cinerea* between hemp varieties and (2) screen organic or soft chemical treatments for the control of *B. cinerea* in IH.

## 2. Results

### 2.1. Detached Leaf Assay Optimisation—Leaf Wounding and Leaf Position

Several preliminary tests were conducted to optimise the inoculation method for detached leaf assays. The factors explored include comparing the effect of wounding leaves prior to inoculation with either fungal spores or a fungal agar plug and the impact of leaf position. Pre-wounding leaves prior to inoculation with a 4 mm fungal plug had no significant effect (*n* = 8, *p* = 0.156, LSD = 2.326) on lesion length (9.09 mm/day) compared to unwounded leaves (7.72 mm/day). Inoculating unwounded leaves by dry brushing conidia had a significant effect (*n* = 6, *p* = 0.042, LSD = 2.948) on lesion length (10.8 mm/day) compared to using the fungal plug (7.72 mm/day; Figure 1A,B). However, the range in average daily lesion length varied considerably between replicates when using dry brushing from the inconsistent quantity of inoculum transfer to the leaf. Given that both methods developed infections, the fungal plug method was selected for subsequent assays to ensure the lesion development would be consistent and not inhibited by the overall leaf length. Control plants without *B. cinerea* did not develop visible lesions (Figure 1C).

There was no significant effect of leaf position on the average lesion length on the first five leaves of a branch (*n* = 40, *p* = 0.076, LSD = 0.5572; Table 1), but the overall length of the leaf was an important factor to consider. Generally, the newest unfurled leaf was short, and the lesion could reach the margins before 6 days if too small. Therefore, to maintain consistency, the second and third newest leaves on any branch were selected for all the subsequent detached leaf assays.

### 2.2. Detached Leaf Assay Screening for Varietal Resistance/Susceptibility

Eight hemp varieties were screened for resistance/susceptibility to *B. cinerea* infection in a series of two experiments. All varieties screened developed lesions following inoculation; however, the variety had a significant effect on average daily lesion length, with SI-1 and Yuma having significantly smaller average daily lesion lengths (Table 2). Leaf size varied amongst varieties, with PUMA generally having longer leaves in the second and third positions of branches. Based on lesion development as an indicator of susceptibility to infection and, importantly, the leaf size of each variety, PUMA was the variety selected for all subsequent detached leaf assays.

### 2.3. Detached Leaf Assay Screening of Active Ingredients

Twenty-two commercially available treatments were screened for efficacy in reducing *B. cinerea* lesion development (Table 3). Nine treatments significantly reduced lesion development compared to the untreated control inoculations: 4 (AI—Potassium Salts of Fatty Acids); 5 (AI—Potassium Bicarbonate + Potassium Silicate + Potassium Salts of Fatty Acids); 8 (AI—Difenoconazole, Thamethoxam, and Abamectin); 9 (AI—Copper Sulphate); 13 (AI—Sodium Bicarbonate); 14, 18, and 20 (AI—Tau-fluvalinate and Myclobutanil); and 17 (AI—Monocalcium Phosphate, Sodium Bicarbonate, and Corn Starch) (Figure 2; Appendix A). Treatment 16 (AI—Prothioconazole, Bixafen, and N, N Dimethyldecanamide) was phytotoxic, causing necrotic spots across the leaf, so it does not have reported efficacy results. Negative figures are shown for treatment 2 (AI—Potassium Bicarbonate and Potassium Silicate; Expt B) and 22 (AI—Copper–Ammonium Complex), resulting from an increased lesion length compared to the untreated control. Based on the percentage reduction in the lesion length compared to the control (Figure 2), treatments 5, 8, 9, and 14 were selected for further evaluation on the effect of adjusting the pH of the treatments prior to application.

### 2.4. Product pH Significantly Affects Product Efficacy

The effect of pH on product efficacy was assessed for treatments 5 (AI—Potassium Bicarbonate + Potassium Silicate + Potassium Salts of Fatty Acids), 8 (AI—Difenoconazole, Thamethoxam, and Abamectin), 9 (AI—Copper Sulphate), and 14 (AI—Tau-fluvalinate and Myclobutanil). Across a range from pH 3 to 10, there was significant variation in the efficacy of treatments in reducing lesion development (Figure 3; Appendix A). Treatment 5 showed improved efficacy at alkaline pH, while treatment 9 showed a reduction in efficacy around neutral pH. Treatment 9 showed excellent efficacy at acidic pH in reducing lesion development but showed symptoms of phytotoxicity at pH 3-pH 7. An additional two experiments using the four treatments 5, 8, 9, and 14 amended to optimal pH were conducted and significantly reduced lesion length, giving an overall product efficacy ranging from an 80% to 89% reduction in lesion growth (Table 4).

### 2.5. Validation of Attached Leaf Assays Using Mature Plants

Detached leaf assays were validated by comparing results to those from equivalent experiments using leaves attached to mature hemp plants. There was no significant difference (*p* = 0.534, LSD = 0.908) between lesion development in the mature plant attached leaf assay and the detached leaf assays. The average lesion length was 7.82 mm/day for the attached leaf assay and 7.04 mm/day for the detached leaf assay (Figure 4). Additionally, there was a similar and significant reduction (*p* < 0.001, LSD = 0.908) in average daily lesion length between the untreated leaves and the leaves treated with Product 14 containing Tau-fluvalinate and Myclobutanil in both the detached and attached inoculation assays.

## 3. Discussion

Fungal pathogens like *B. cinerea* are a regular problem within MC protected cropping systems, resulting in yield and product quality losses. There is a need to rapidly screen for varietal susceptibility as well as product effectiveness in reducing disease symptoms in this emerging industry. The validation of a detached leaf assay method allows for the rapid screening of varieties or products with high replication and consistency, without the risk of contamination or loss of stock plants.

The application of the pathogen as a mycelium contained on an agar plug onto unwounded leaves was determined as the most effective method to obtain reproducible results in this study. A previous study considering *B. cinerea* infection of strawberries using a similar detached leaf system wounded leaves prior to infection [13]. Wounding would be expected to increase susceptibility to necrotrophic pathogens, such as *B. cinerea*, as it creates access to plant nutrients and damaged necrotic tissue; however, other studies have demonstrated that wounding is also able to induce defences against pathogens and provides a degree of resistance [14]. This may explain the lack of significant difference in lesion development between wounded and unwounded leaves in our experiments.

These results correspond well with whole-plant assays and are consistent with previous studies using detached leaf assays [9,10,11,12]. In addition to the technical and practical advantages of the detached leaf assays over whole plant assays, the use of a detached leaf also eliminates the systemic effects of treatments if more than one leaf per whole plant is tested. Pathogen infection at any one site on a plant can trigger systemic acquired resistance (SAR), reducing susceptibility to subsequent infections [15], although *B. cinerea* infection may be an exception to inducing SAR [16]. In addition to systemic responses within the plant to pathogen attack, some product treatments are similarly able to operate systemically and may confound results; therefore, a study specifically investigating the systemic effects of a product treatment will not be aided by a detached leaf assay, so the choice of assay should be determined by the information required.

While the PUMA variety of IH was selected for detached leaf assays within this study, all varieties tested were vulnerable to *B. cinerea* infection with no one variety showing obviously superior resistance to lesion growth. Glasshouse-grown MC and IH plants have a high degree of susceptibility to a variety of bud and leaf rot pathogens, including *B. cinerea*, *Penicillium* and *Fusarium* species, *A. alternata*, and powdery mildew [2,17]. The development of disease-resistant cultivars would greatly improve disease management in intensive MC production. The detached leaf assay is a fast, cost-effective method to screen existing and new cultivars and crosses between varieties before growing them on a large scale within growing facilities. Susceptible varieties or crosses can be eliminated from the breeding program before spending time, space, and money on susceptible lines. Due to the European Pharmacopoeia standards surrounding chemical residues on MC products [18], the use of many commercial fungicides is precluded in MC production, particularly in the direct treatment of inflorescence. The Australian Therapeutic Goods Order outlines the quality standards of MC administered in Australia [19]. Within Australia, MC is still regarded as an emerging industry, with no products registered with the Australian Pesticides and Veterinary Medicines Authority. In some cases, gamma irradiation is used to reduce microbial contamination to pharmaceutically acceptable levels [20], further substantially adding to the cost of production. Growing disease-resistant varieties is, therefore, by far, the best management strategy for disease control.

In this study, we screened a variety of chemical or organic active ingredients for efficacy against *B. cinerea* lesion development. The most consistent reduction in lesion growth was obtained from Treatment 14, which contains Tau-fluvalinate and Myclobutanil as active ingredients. Treatments 18 and 20 both contained the same active ingredients and also significantly reduced lesion growth. Tau-fluvalinate is a synthetic pyrethroid insecticide most notable for its use in varroa mite control in the honey industry [21]. Myclobutanil is a sterol-biosynthesis-inhibiting fungicide shown to inhibit Botrytis mycelium growth [22]. Both ingredients can have mild to moderate toxic effects on humans or non-target organisms and, therefore, may only be appropriate for foliar control of *B. cinerea* and other pathogens well prior to inflorescence harvest. Formulations containing only myclobutanil may be more appropriate for *B. cinerea* control to reduce chemical residues, although mites are also a persistent issue in intensive MC production. The development of resistance to insecticides or fungicides such as these remains difficult [23]; therefore, treatment with products should only be used as part of a larger integrated disease management program.

Additionally, control of *B. cinerea* was also accomplished using softer chemical options, including Treatment 5, which contained a number of potassium salts. Products containing these active ingredients present a promising alternative for the control of bud rot in MC. Potassium salts, including potassium carbonate and salts of fatty acids, have been previously shown to be effective in reducing *B. cinerea* growth [24,25,26]. Interestingly, while there are several studies showing excellent control of *B. cinerea* with Trichoderma species [27,28], in our study, we did not see a significant inhibition of *B. cinerea* lesion development with Trichoderma treatment.

Many fungi, including *B. cinerea*, grow better in acidic to neutral pH environments than in alkaline environments [29]. Products like Treatment 5 are naturally highly alkaline when dissolved in water, raising the possibility that their efficacy is tied merely to the change in pH. Previous research has refuted this, suggesting that while high pH can inhibit fungal growth, it is additive to the effect of the potassium salts [30,31]. This is reflected in our results, which showed efficacy for Treatment 5 at each pH tested but with higher efficacy at alkaline pH. Likewise, the observed phytotoxicity of the copper-containing Treatment 9 at acidic pH is in line with previous findings, owing to greater copper solubility at low pH [32]. This highlights the need to apply individual treatments at optimal pH to effectively reduce lesion development.

Overall, the use of this optimised detached leaf assay facilitated the rapid screening of IH varietal susceptibility to *B. cinerea* and identified promising treatment options with strong correlations to whole-plant trials. Regarding treatment efficacy, Treatment 14 (containing Tau-fluvalinate and Myclobutanil) consistently reduced lesion growth. Additionally, softer chemical options, such as those containing potassium salts, showed effectiveness in *B. cinerea* control, offering a potential alternative for bud rot management in MC. The results of this study also shed light on the role of pH in treatment efficacy, highlighting the need to apply treatments at optimal pH levels for maximum effectiveness. The use of the detached leaf assay methodology holds promise for higher throughput screening of new varieties and breeding lines for pathogen resistance, as well as determining effective treatment options for pathogen control.

## 4. Materials and Methods

### 4.1. Origin and Maintenance of Pathogen Isolate

The single-spore *B. cinerea* isolate DAR83297 (Living Culture Collection at NSW Department of Primary Industries, Orange, NSW, Australia, 2020) used for these experiments was sub-cultured onto 25% potato dextrose agar (PDA) plus Novobiocin (Sigma, Darmstadt, Germany) and incubated at 23 °C for 14 days. The 25% PDA plus Novobiocin medium was prepared with 9.75 g/L PDA (BD Difco) and 11.25 g/L base agar (Gelita Grade J3) in distilled water (1 L) pre-autoclave. Post-autoclave, Novobiocin stock solution (0.0175 g/mL) was added at 6.65 mL/L. *B. cinerea* plugs at 4 mm in diameter were harvested from the outside growing edge of actively growing cultures and placed in the centre of 90 mm Petri plates containing 20 mL of fresh PDA plus Novobiocin. All petri plates were kept at 23 °C in the dark for a total of six days before being used in a detached leaf inoculation assay.

### 4.2. Growth and Maintenance of Plant Material

Low-THC *Cannabis sativa* varieties SI-1, YUMA, BAMA, PUMA, Han NW, Han NE, Han FNH, and Han FNQ were supplied by Southern Cross University, Lismore, NSW, and used in this study under NSW Government hemp license number 52775. All plants were potted in 3 L pots filled with Searles Premium Potting Mix. Plants were grown in a glasshouse under a temperature of 28 °C (daytime) to 18 °C (night-time) with a supplementary light intensity of 300 µmol m^−2^ s^−1^ and a photoperiod of 18/6 h (light/dark). This photoperiod ensured that the plants remained in a vegetative growth phase and did not transition to flowering. Plants were pruned every two weeks to manage canopy size and promote new growth. Plants were fertilised once every two weeks using Amgrow Nitrosol Liquid Plant Food (40 mL/9 L) (Amgrow Australia Pty Ltd., Rhodes, NSW, Australia), a nutrient solution containing nitrogen, phosphorus, and potassium (12-2-13).

### 4.3. Optimisation of Leaf Inoculation for Detached Leaf Assays

Leaves were removed from the plants and transported from the glasshouse to the laboratory in a Styrofoam container with a wet paper towel at the base to keep the leaves fresh. Clean 500 mL plastic takeaway containers were pre-labelled with each treatment and prepared by placing a piece of paper towel (VIVA select a size) folded in half in the base of the container, with 5 mL of sterile water dispensed onto the paper towel using a pipette. Leaves were placed in individual containers adaxial side up.

To optimise infections, leaves were wounded using a modified method described by [14] by gently pressing the detached leaf onto sandpaper of medium grit (P120) or left unwounded. The wounded area was marked with a white marking pen before either Botrytis spores from a 6-day-old culture were dry brushed on the wounded area using a fine paintbrush (Figure 1A) or a 4 mm *B. cinerea* fungal plug was cut from the actively growing edge of an inoculated plate using a 4 mm cork borer and placed in the centre of the wounded area. The 4 mm *B. cinerea* fungal plug was inverted and placed on the middle fan leaf along the centre vein (Figure 1B). The 4 mm PDA plugs were used as controls (Figure 1C). Whilst both methods caused infection, the fungal plug inoculation method provided repeatable infection results with minimal variation between replicates and was used in all subsequent detached leaf assays.

The potential for leaf position to affect lesion development was assessed. Five leaves from the top of each branch (1 = newest leaf, 5 = oldest leaf) were selected and placed in individually labelled containers adaxial side up and inoculated using the fungal plug method described above.

### 4.4. Detached Leaf Assay Screening for Varietal Susceptibility

Eight hemp varieties were screened for their resistance/susceptibility to infection by *B. cinerea* using the detached leaf assay. Five detached leaves of each variety were inoculated with a 4 mm *B. cinerea* fungal plug, as well as five controls that had a 4 mm PDA plug without inoculation, as described above. This experiment was replicated twice.

### 4.5. Detached Leaf Assay Screening of Active Ingredients

Twenty-two products with active ingredients (Table 3) were selected as treatments based on a literature search for fungicidal products with a short withholding period. Each of the treatments (that were not premixed) were applied at label rate using sterile water.

Detached leaves were inoculated using the method described above before 0.2 mL of each treatment was sprayed on each leaf, from approximately 20 to 30 cm above, using a Mad™ Intranasal Mucosal Atomization Device (Teleflex^®^, 3015 Carrington Mill Boulevard, Morrisville, NC, USA, 27560). Lids were put on the 500 mL plastic containers and then incubated in the dark at 23 °C. After 6 days, disease development was assessed by measuring the length of diseased lesions that had spread beyond the mycelial plugs. Average daily lesion length was calculated as follows: total length (mm)—4 mm fungal plug/6 days of growth. In a small percentage of experiments (<8%), agar plugs fell off the leaf surface or failed to develop lesions. These replicates were excluded from calculations. PDA plug inoculations (controls) did not produce lesions.

Four treatments with high efficacy in the screening assays were then assessed for the effect of adjusting the pH on lesion development. Premixed treatments were decanted into 50 mL sample containers. All other treatments were mixed at label rate with sterile water and decanted into 50 mL sample containers. The pH was measured using a TPS Aqua pH meter (Unit 1, 8 Bult Drive, Brendale, QLD, Australia, 4500) before adjusting to pH 3, 4, 5, 6, 7, 8, 9, and 10 using “Manutec pH Raise” or “Manutec pH Lower” (30 Jonal Drive, Cavan, SA, Australia, 5094). The pH-adjusted treatments were applied to the detached leaves and incubated in the dark, as described above.

### 4.6. Attached Leaf Assays Using Mature Plants

To validate the results observed in the detached leaf assays, lesion development on attached leaves was assessed using mature plants. Six mature IH plants in 4 L pots filled with Searles Complete Potting Mix were used. Two plants were designated for the *B. cinerea* (untreated), two for the treated (*B. cinerea* plus—Treatment 14), and two for PDA plugs only. Ten leaves per plant were inoculated using a 4 mm PDA plug for control inoculation or a 4 mm *B. cinerea* fungal plug for the Botrytis inoculation with the plugs held in place using a sterilised pin (Figure 5A). A volume of 0.2 mL of Treatment 14 was sprayed on 20 inoculated leaves, from approximately 20 to 30 cm above, using a Teleflex^®^ Intranasal Mucosal Atomization Device. Ziplock sealable sandwich bags (22 cm × 22.5 cm) were placed over the attached leaf and sealed to keep the fungal plugs from drying out (Figure 5B). Whole plants were placed in a propagating room under a 16/8 h light regime at 23 °C. After 6 days, disease progress was assessed by measuring the length of diseased lesions that had spread beyond the mycelial plugs. Average daily lesion length was calculated as follows: total length (mm)—4 mm fungal plug/6 days of growth. Concurrently, detached leaves were also inoculated using the detached leaf assay screening of active ingredient method described earlier to compare them with the attached leaf assay results.

### 4.7. Statistical Analyses

Statistical differences between treatments were determined using ANOVA in GenStat 21st Edition (VSN International, Hertfordshire, UK). A *p*-value of less than 0.05 was considered significant when comparing the least significant difference (LSD) of means at a 5% level using a Bonferroni test.

## Figures and Tables

**Figure 1 plants-12-03278-f001:**
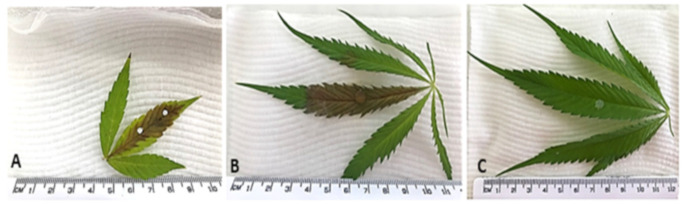
Detached hemp leaf assays. (**A**) Hemp leaf inoculated after wounding with sandpaper in area marked with a white pen by dry brushing *Botrytis cinerea* spores between the marked area. After 6 days, the lesion had spread beyond the inoculation site. (**B**) Leaf lesion spreading from the 4 mm *B. cinerea* fungal plug on the middle fan leaf from a hemp plant. (**C**) No leaf lesion spreading from the 4 mm PDA plug (control) on the middle fan leaf from a hemp plant.

**Figure 2 plants-12-03278-f002:**
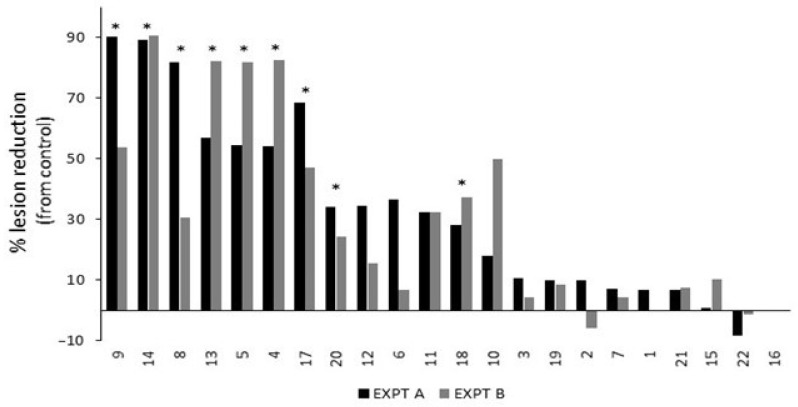
Percentage lesion reduction results from a series of repeated (EXPT A and EXPT B) detached leaf assay experiments for commercially available and common natural treatments, as listed in Table 3. Asterisks indicate significant reduction (*p* < 0.05) in lesion length from untreated control *B. cinerea* inoculations across both experiments.

**Figure 3 plants-12-03278-f003:**
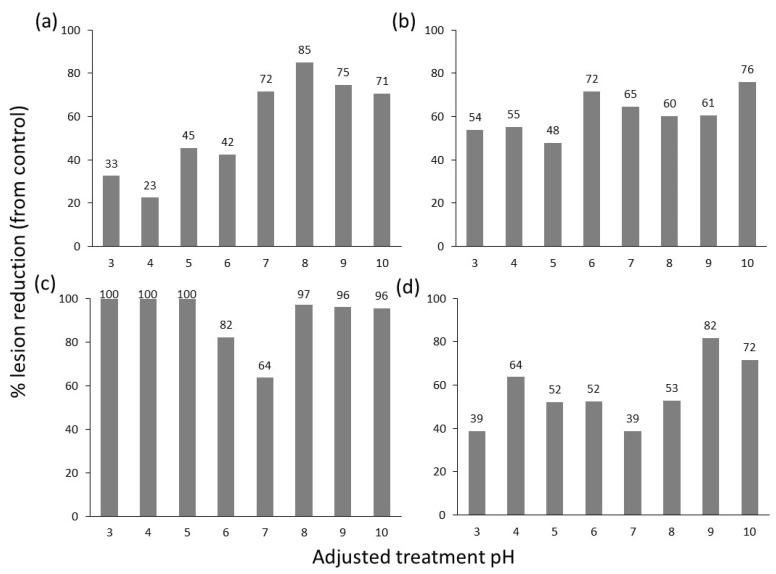
Summary of the average of two repeated detached leaf assay experiments showing the percentage reduction in lesion length following the application of treatments 5 (**a**), 8 (**b**), 9 (**c**), and 14 (**d**) with the pH adjusted to pH 3–10.

**Figure 4 plants-12-03278-f004:**
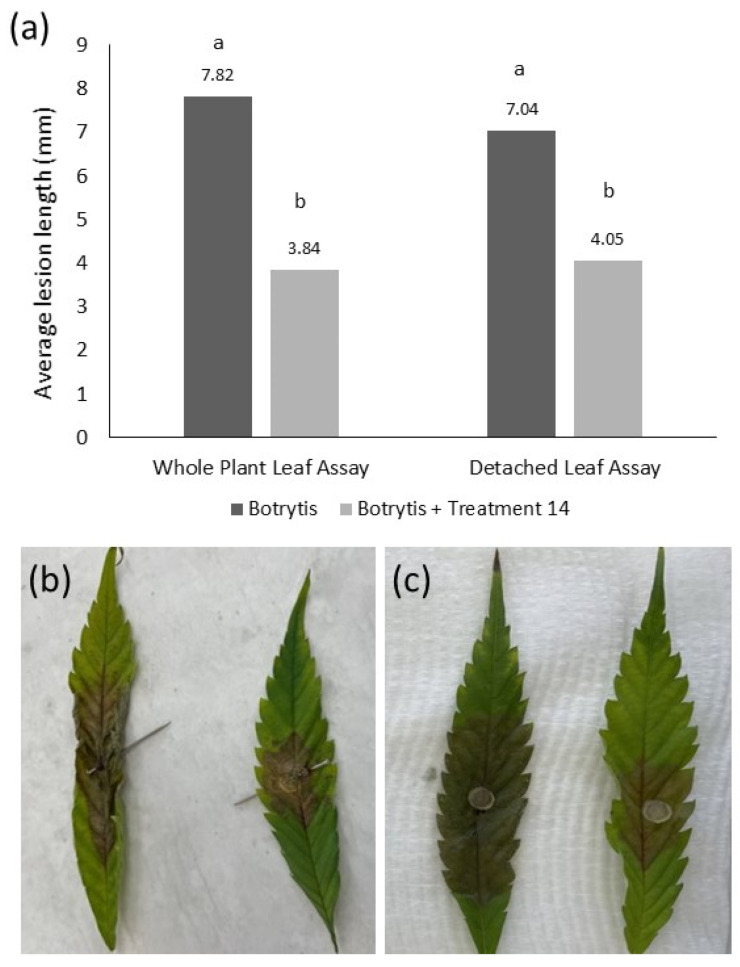
Validation of the detached leaf assay using leaves on mature whole plants. (**a**) Results show no significant difference in assay type but a significant effect of treatment on average lesion length caused by *B. cinerea*. Different letters above bars represent significant differences between treatments (*p* < 0.05). (**b**) Attached leaves inoculated with *B. cinerea* only (left) or *B. cinerea* with Treatment 14 (right). (**c**) Detached leaves inoculated with *B. cinerea* only (left) or *B. cinerea* with Treatment 14 (right).

**Figure 5 plants-12-03278-f005:**
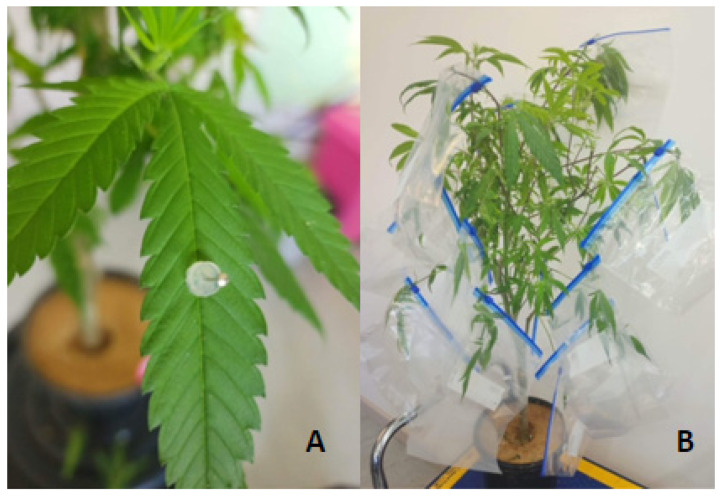
Inoculation of attached leaves on mature plants. (**A**) Inoculated attached leaf with Botrytis fungal plug secured using a sterilised pin. (**B**) Leaves enclosed in a sealable plastic bag.

**Table 1 plants-12-03278-t001:** Effect of leaf position on average lesion length inoculated with a 4 mm Botrytis fungal plug. Same letters indicate no significant difference (*p* < 0.05) according to the least significant difference test.

Leaf Position	Average Lesion Length (mm/day)
1	5.875 a (range, 5.4–6.7)
2	5.975 a (range, 4.5–7.0)
3	6.337 a (range, 5.0–7.5)
4	6.225 a (range, 5.0–7.5)
5	5.587 a (range, 4.5–6.6)
*p*-value	0.076
LSD	0.5572
Replicates (*n*)	40

**Table 2 plants-12-03278-t002:** Susceptibility to *Botrytis cinerea* infection differs amongst hemp varieties. Different letters indicate significant difference (*p* < 0.05) according to the least significant difference test.

Hemp Variety	Average Lesion Length (mm/day)	
	Experiment A	Experiment B
SI-1	9.78 a	9.00 b
YUMA	10.42 a	7.89 a
HAN NW	10.49 b	9.80 c
BAMA	10.64 b	9.01 b
HAN NE	10.67 b	10.62 c
PUMA	10.69 b	10.54 c
HAN FNH	10.80 b	9.61 c
HAN FNQ	11.83 c	10.62 c
*p*-value	<0.001	<0.001
LSD	0.661	0.990
Replicates (*n*)	20	20

**Table 3 plants-12-03278-t003:** Product treatments and active ingredients screened for efficacy against *Botrytis cinerea* using the detached leaf assay. All treatments were prepared at the given label rate.

Treatment No.	Active Ingredients (AIs)	Label Rate
1	*Trichoderma harzianum*	1 g/700 mL
2	633 g/kg Potassium Bicarbonate, 312 g/kg Potassium Silicate	4 g/L
3	940 g/kg Potassium Bicarbonate	3.8 g/L
4	194 g/L Potassium Salts of Fatty Acids	200 mL/L
5	633 g/kg Potassium Bicarbonate, 312 g/kg Potassium Silicate + 194 g/L Potassium Salts of Fatty Acids	4 g/L and 200 mL/L
6	10% Soybean Oil, 5% Corn Oil, 85% Water, Guar Gum, Glycerine, Citric Acid, Soap, Vanillin	57 g/4.5 L
7	0021% *w*/*v* Available Chlorine	Premixed
8	0.167 g/L Difenoconazole, 0.100 g/L Thamethoxam, 0.015 g/L Abamectin	Premixed
9	25% *w*/*w* Copper as Sulphate, 12% Sulphur as Sulphate	10 g/L
10	Apple Cider Vinegar	10 mL/L
11	3% Hydrogen Peroxide	Premixed
12	11.82 g/L Azadirachtin A and B present as 29.55 g/L Azadirachta indica extract	10 mL/5 L
13	100% Sodium Bicarbonate	1 g/L
14	1 g/L Tau-fluvalinate, 0.05 g/L Myclobutanil	Premixed
15	5 g/L Spinetoram in the form of a suspension concentrate	5 mL/L
16	150 g/L Prothioconazole, 75 g/L Bixafen, 523 g/L N, N Dimethyldecanamide	0.9 mL/L
17	Monocalcium Phosphate, Sodium Bicarbonate, and Corn Starch (Baking powder)	Premixed
18	9.0 g/L Tau-Fluvalinate, 4.4 g/L Myclobutanil	7.5 mL/L
19	100% Citric Acid	1 g/L
20	0.1 g/L Tau-fluvalinate, 0.05 g/L Myclobutanil	Premixed
21	10–30% *w*/*w* Phosphoric Acid, Monopotassium Salt	6 mL/L
22	93 g/L Copper–Ammonium Complex	10 g/L

**Table 4 plants-12-03278-t004:** Average per cent reduction in lesion development following *B. cinerea* inoculation for product treatments at optimal pH compared to untreated controls across a total of four experiments (*n* = 20 per experiment) with standard error (SE).

Treatment No.	Optimal pH	Average Percent Lesion Reduction	SE
5	8	83.5	6.1
8	10	79.8	5.5
9	9	89.1	6.4
14	9	83.5	4.3

## Data Availability

The data presented in this study are contained within the manuscript or Appendix A.

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
