# Peer review of "Using Detached Industrial Hemp Leaf Inoculation Assays to Screen for Varietal Susceptibility and Product Efficacy on *Botrytis cinerea"

_plants, 2023, doi:10.3390/plants12183278_

Round 1

Reviewer 1 Report

The manuscript is well written and the methodology used is appropriate for the aim proposed. The results are clear and trustable and the conclusions are in accordance with the data presented. However, the manuscript lacks in novelty and originality, describing after all just the suitability and reliability of a detached leaf assay for differentiating varietal susceptibility and  screening treatments for disease caused by Botrytis cinerea. Working with medicinal plants and pathogens that can undirectly affect also human helath, it would be more interesting to have reports on treatments with no side effects on human beings. I would suggest to submit the manuscript to journals more focused on methods.

The english language is appropriate.

Author Response

Reviewer   1

Thank you for your review of this manuscript and your suggestions.

Working with medicinal plants and pathogens that can indirectly affect also human health, it would be more interesting to have reports on treatments with no side effects on human beings. We believe our findings in regard to products with potassium salts addresses this issue.  Although this paper is predominantly a methodology paper, we believe the findings in this manuscript fit the scope of this journal and the submitted section Plant Protection and Biotic Interactions. We will continue with the submission process to this journal, following minor revisions. 

Reviewer 2 Report

Dear Authors,

Manuscript ID: plants-2595122 entitled “Using detached industrial hemp leaf inoculation assays to screen for varietal susceptibility and product efficacy on Botrytis cinerea” is a well-written manuscript, the methods applied are by the expected results and the experiments are well conducted.

The manuscript needs some minor revision before publication and I have attached the manuscript with my comments.

Author Response

Reviewer 2

Thank you for your review of this manuscript.    We have made all changes suggested by the reviewer. Point by point response to each of the reviewers' comments are below:

Line 11 - Inserted Pers.

Line 13 - Changed fungicide to fungicides

Line 15 - Inserted hyphen (time-consuming)

Line 17 - Deleted "a"

Line 22 - Changed disease to diseases

Line 44 - Added Pers.  and (Fr.) Keissl

Line 51 - Changed Botrytis cinerea to B. cinerea

Line 54 - Changed increases to increase

Line 165 - Changed Average to The average

Line 195 - Deleted "the"

Reviewer 3 Report

After careful evaluation of the manuscript title "Using detached industrial hemp leaf inoculation assays to screen for varietal susceptibility and product efficacy on Botrytis cinerea", you can find below my comments. 

- The manuscript is well written, understandable to the reader, and appropriate to the requirements of the Plants journal.

- The purpose of the manuscript is well formulated.

- Research results are well presented and described. They complement the knowledge for scientists dealing with the health of plants with health-promoting properties.

- Figures (5) and Table (4) are correct.

- The presented results are of great importance and affect the progress in a given field of research.

- Discussion is well conducted with the use of relevant literature sources.

- Materials and Methods are sufficiently described.

- Due to the growing global interest in plants with health-promoting properties, there is a high probability that the manuscript will be cited by researchers dealing with the health issues of Cannabis sativa in the future, especially in the direction of reducing the occurrence of Botrytis cinerea.

I would like to propose some corrections:

2. Results

- Lines 86, 88 - please remove the space after the = sign.

- Please increase the sharpness of the elements of Figure 1.

4. Conclusions

- Please write synthetic conclusions from your research.

Best regards

Author Response

Reviewer 3

Thank you for your review of this manuscript.    We have made all changes suggested by the reviewer. Point by point response to each of the reviewers' comments are below:

Line 86 - Removed space after =

Line 88 - Removed space after =

Figure 1 - increased sharpness of elements in Figure 1

Write a synthetic conclusion - Last paragraph in discussion expanded to draw overall conclusions from the multiple experiments reported in this manuscript.

Round 2

Reviewer 1 Report

In the answer to revisions, the authors claimed that they believe that the findings with the use of potassium products are addressing the issue about novelty and soundness of the manuscript. If it is so, I would suggest to stress more this part in the final discussion and remarks, where they actually improuved the conclusions following other suggestions. I would suggest to rewrite the coclusive part of the abstract as well, stressing this part, in order to render the manuscript less technical and more of a wide interest for scientists. 

Author Response

Thank you for you feedback. We believe we have addressed the findings of this research in the discussion, particularly in paragraphs starting on line 229 and 243. We did rewrite/update the conclusive section of the abstract to reflect your comments.
